# Susceptibility of Different Life Stages of Kudzu Bug *Megacopta cribraria* (F.) (Hemiptera: Plataspidae) to Two Different Native Strains of *Beauveria bassiana*

**DOI:** 10.3390/pathogens11091028

**Published:** 2022-09-09

**Authors:** James Paul Glover, Maribel Portilla, Katherine Parys, Clint Allen, Justin George, Gadi V. P. Reddy

**Affiliations:** 1Southern Insect Management Research Unit, United State Department of Agriculture-Agricultural Research Service, Stoneville, MS 38776, USA; 2Pollinator Health in South Crop Ecosystems Research Unit, United State Department of Agriculture-Agricultural Service, Stoneville, MS 38776, USA

**Keywords:** kudzu bug, entomopathogenic fungi, lethal concentration 50, *Beauveria bassiana*, agricultural pest

## Abstract

This is the first study that examined and compared the survival, LC_50_, and RR_50_ estimates of *Megacopta* *cribraria* F. (Hemiptera: Plataspidae) nymphs and adults that were exposed to two native *Beauveria* *bassiana* isolates (Previously codified as NI8 and KUDSC strains) at four concentrations. The greatest reduction in survival and mortality was observed primarily on or after 10 d post-exposure to *B. bassiana* isolates. Survival of early instars (2nd, 3rd) were not affected by either strains or concentration at 3 d and 5 d post-exposure. Survival of later instars (5th) and adults was significantly reduced when exposed to the KUDSC strain at all concentrations. Comparison of dose–mortality values (LC_50_) using resistance ratios (RR_50_) were significantly different between life stages of the kudzu bug for both strains of *B. bassiana*. The LC_50_ values showed that kudzu bug adults are more susceptible than any other life stage when exposed to either strain. The KUDSC strain was more pathogenic than NI8 10 d after exposure, but NI8 exhibited significantly higher pathogenicity than KUDSC 20 d after exposure. Our results suggest potential field application of *B. bassiana* for kudzu bug control and their integration into pest management strategies to suppress them before they cause economic damage to soybean crops.

## 1. Introduction

Kudzu bug, *Megacopta cribraria* (F.) (Hemiptera: Plataspidae), is an introduced urban and agricultural pest of soybeans (*Glycine max* (L.) Merrill) (Fabales: Fabaceae) grown in the southeastern United States. Native to southeastern Asia, kudzu bugs were first identified in the United States as an urban pest in northeastern Georgia in 2009, and one of the two representatives of the hemipteran family Plataspidae introduced to the Americas [1,2]. The geographic range of kudzu bugs expanded rapidly within the United States, spreading as far west as Louisiana and north to Maryland by 2015, occupying 652 counties in the United States by 2017, and west into Texas by 2021 [3,4,5]. Models indicate that much of Central America, South America, and the eastern United States are susceptible to future invasion by this species [6], but thermal limits should restrict establishment north of the state of Maryland [7]. Assumed to have established from a single introductory location in Georgia, kudzu bugs also vary in size and flight potential in relation to the distance from the original introduction [8,9].

Feeding by kudzu bugs can cause yield losses in untreated soybean across the southeastern United States [10]. Kudzu bugs, both adults and nymphs, feed on kudzu variety *lobata* (*Pueraria montana* (Lour.) Merr. (Fabales: Fabaceae) (Willd), as the name implies, along with soybeans and a variety of other legumes [1,11,12]. Adults overwinter in the leaf litter in woody or forested areas, moving into the litter in late November and early December, and are often observed aggregating on the outside walls of houses and other urban structures [7,13]. The species is bivoltine in the United States, and both generations have been observed using kudzu and soybean as host plants [14,15,16]. Kudzu bugs feeding on soybean are capable of causing severe reductions in yield, up to 60% [15]. Options for chemical control are available for both in-field and structural use [17,18].

Organic bean growers in the southeastern United States may have severely damaged or destroyed crops by and have limited control options for kudzu bugs [19]. Only a few natural enemies of kudzu bugs have been found in the United States [20,21,22,23]. The pathogenic fungus *Beauveria bassiana* (Bals.-Criv.) Vuill. (Cordycipitaceae) has been observed to cause epizootic outbreaks in kudzu bug populations across the region [10,19,24,25,26,27].

In 2005, a native Mississippi Delta strain of *B. bassiana* (NI8) was isolated from *Lygus lineolaris* Palisot de Beauvois (Hemiptera: Miridae) and was effective in measurable pest suppression of *L. lineolaris* [28]. A native South Carolina strain of *B. bassiana* originally isolated from kudzu bugs was identified and codified as KUDSC in 2015 and successfully established and colonized insects [25]. This study is the first investigation that examined and compared the survival, mortality, dose–mortality values (LC_50_), and resistance ratio (RR_50_) estimates of nymphs and adults of *M. cribraria* exposed to four concentrations of both NI8 and KUDSC native strains of *B. bassiana*.

## 2. Materials and Methods

### 2.1. Kudzu Bug Collections

Kudzu bugs of all life stages were collected during June of 2015 from kudzu plants in Yazoo County, located south of the town of Yazoo City, Mississippi. Insects were collected using a 15-inch diameter sweep net from large patches of kudzu on the roadside, and individuals were returned to the laboratory at the USDA-ARS in Stoneville, Mississippi. After collection, insects were sorted by instar and maintained overnight at 27 °C in screened insect observation cages (Popup Rearing Cages 1466 PB) (30 × 30 × 30 cm, Bugdorm^TM^, BioQuip Products, Rancho Dominguez, CA, USA) containing kudzu plants for food until experiments were conducted.

### 2.2. B. bassiana Spore Preparation

A strain of *B. bassiana* (NI8), originally isolated from *L. lineolaris* and native to the Mississippi Delta [28], is regularly cultured at the USDA-ARS Southern Insect Management Research Unit in Stoneville, Mississippi. Strain KUDSC was originally isolated from kudzu bugs collected at the Clemson University Edisto Research and Education Center in Blackville, South Carolina, SC, USA. [25]. Technical grade powder of both strains was produced using a medium-scale biphasic culture system for solid-substrate fermentation [25,29].

### 2.3. Bioassay Procedures

Bioassays were conducted using both native strains of *B. bassiana* on four nymphal instars (2nd–5th) and kudzu bug adults. At first, instar nymphs were not included in this study due to high handling losses during field collections. Detailed methodology for spore germination and calculation for spores/mL and spores/mm^2^ have been published previously [25,29,30,31]. Each treatment suspension for both isolates (7 × 10^4^, 7 × 10^5^, 7 × 10^6^, 7 × 10^7^ spores/mL) (NI8: 380, 38, 3.8, 0.38 spores/mm^2^) (KUDSC: 350, 35, 3.5, 0.35 spores/mm^2^) and water control were applied to each of three replicated groups of 30 kudzu bugs, for a total of 900 individuals of each specific age category, and an overall total of 2700 individual of kudzu bugs. For each replicated group, treatments were applied in a 6 mL spray volume from the lowest concentration to highest concentration, and nozzles were changed between each treatment to avoid cross-contamination. As described in [31], conidia suspencions were applied with specially designed spray tower equipped with an air atomizing nozzle ¼ J, fluid cap 2850, and air cap 70 (FN5925-001-001A, Wheaton, IL, USA). After each application, the cohort of kudzu bugs was released into an observation insect cage. To allow insects to dry post-spray, a single cage was utilized for each replicate and treatment combination. Once dry, insects that had been sprayed were placed individually into plastic cups containing a solid diet developed for plant-feeding hemipterans [31,32]. Following the spray exposure, kudzu bugs were held for 20 d in an environmental chamber at 27 °C, 65% R.H., with a photoperiod of 12:12 (L:D) h. Insects were considered dead when no movement was observed after prodding. Dead insects were kept individually in their original cup and checked daily for sporulation of *B. bassiana* fungus on the cadaver.

### 2.4. Statistical Analyses

Experiments were analyzed as a randomized complete block with the factorial arrangement in PROC GLIMMIX [33]. Nine treatments at each life stage examined included *B. bassiana* strains NI8 and KUDSC, and the water control. Each treatment combination was repeated three times. Nonparametric estimates of survival were compared between treatments using PROC LIFETEST. Mortality was corrected for control effects by using Abbot’s formula [34]. Slopes, LC_50_, and RR_50_ estimates of *B. bassiana* strains were obtained from the corrected data from bioassays using PROC PROBIT [33].

## 3. Results

### 3.1. Nymph and Adult Survival and Sporulation

Significant differences were observed in the number of days until death after exposure for all kudzu bug developmental stages. There were statistically significant differences in all life stages among concentrations for either strain (−2log-rank test of equality over strata) (Figure 1, Figure 2 and Figure 3). Survival was not affected by either of the *B. bassiana* isolates or concentrations assayed for any life stage in this experiment one day post-exposure except for adults treated with the KUDSC strain. Regardless of treatment combination, differences in survival never increased from day 15 through day 20 in this study (Figure 1, Figure 2 and Figure 3). The greatest reduction in survival was observed primarily on or after 10 d post-exposure to the *B. bassiana* isolates. Survival of 2nd and 3rd instars were not affected by either strain or concentration investigated at 3 d and 5 d post-exposure (Figure 1). Survival of 5th instar (Figure 2D) and adults (Figure 3B) was significantly reduced when exposed to the KUDSC strain at all concentrations. No mortality of late instars or adults were observed with the NI8 strain 5 d after exposure. Little variation was observed in the average number of days for cadavers to sporulate irrespective of isolate and concentration, ranging from 2 to 3 d for nymphal stages and 2 to 5 d for adults.

### 3.2. Nymphal and Adult Mortality

#### 3.2.1. Second and Third Instars of Kudzu Bug

A significantly different strain–concentration interaction was observed on day 5 for 2nd instar nymphs (*F* = 2.03; df = 8, 809; *p* < 0.0281). No significant interaction was observed for 3rd instar nymphs on day 5 (*p* < 0.05). Unexpected early mortality was observed in 2nd instar bugs exposed to isolate KUDSC strain at the lowest concentration (7 × 10^4^) 5 d after exposure (22.22 ± 1.56) (Figure 1B). That mortality was obtained two days before the higher concentrations (7 × 10^6^ and 7 × 10^7^) for both strains (Figure 1A,B). However, mortality values remained low for the next 5 d, with a maximum mortality of 37.7 ± 5.1 to day 20, with no significant differences compared with the KUDSC 7 × 10^5^ concentration (37.0 ± 5.1) (Figure 1B/Table 1). Mortality of 2nd and 3rd instars with higher concentration started at day 5 for NI8 and day 6 for KUDSC and increased rapidly to day 15, and their mortality values for both nymphal stages were evidently higher with NI8 than KUDSC (Figure 1). Significant differences in mortality were observed for both instars at day 10 (2nd instar: *F* = 13.35; df = 8, 809; *p* < 0.0001) (3rd instar: *F* = 9.25; df = 8, 809; *p* < 0.0001) and 20 d (2nd Instar: *F* = 13.69; df = 8, 809; *p* < 0.0001) (3rd instar: *F* = 17.13; df = 8, 809; *p* < 0.0001) after exposure (Table 1).

#### 3.2.2. Fourth and Fifth Instars of Kudzu Bug

Similar to the 2nd instars, 4th instar nymphs were not significantly affected by *B. bassiana* isolate or concentration on day 5 after exposure (*F* = 1.00; df = 8, 809; *p* < 0.4721) (Figure 2A,B). However, there were statistically significant differences in mortality when strains and concentrations were compared on day 10 (*F* = 3.22; df = 8, 809; *p* < 0.0001) and 20 (*F* = 6.76; df = 8, 809; *p* < 0.0001). The mortality trend for 4th and 5th instars exposed to both isolates had similar interaction for all concentrations. The mortality value for 4th instar nymphs slightly increased when bugs were exposed to the KUDSC strain with the highest concentration 20 d after exposure, with 85.56 ± 3.73% and 95.56 ± 2.18% for NI8 and KUDSC, respectively. Mortalities for 5th instars slightly increased when the bugs were exposed to NI8 with the highest concentration 20 d after exposure with 72.22 ± 5.28% and 66.67 ± 4.50% for NI8 and KUDSC, respectively, showing highly significant differences among strains (Figure 2C,D) (Table 1). Although 5th instars were highly susceptible to the KUDSC strain, showing mortality from day 1 (Figure 1D), mortality values were not higher than those observed for 2nd (73.33 ± 4.70%), 3rd (85.65 ± 3.71%), and 4th (95.56 ± 2.18%) instars 20 d after exposure. The 4th and 5th instars exposed to the lowest concentration (7 × 10^4^, 7 × 10^5^) did not significantly differ from those exposed to the water control (Table 2). Significant differences in mortality were observed for the 5th instar at day 5 (*F* = 28.05; df = 8, 809; *p* < 0.0001), 10 (*F* = 9.29; df = 8, 809; *p* < 0.0001), and 20 d (*F* = 11.63; df = 8, 809; *p* < 0.0001) post-exposure (Table 1).

#### 3.2.3. Adults of Kudzu Bug

The mortality trend of kudzu bug adults differed from nymphal stages when exposed to NI8 and KUDSC strains. No significant differences were found among lower concentrations of NI8 (7 × 10^4^, 7 × 10^5^, 7 × 10^6^), and no significant differences were observed with any concentrations of the KUDSC strain (Figure 3A,B). Similar to 5th instars, adult mortality started 1 d after the insects were exposed to any concentration of the KUDSC strain. However, at the end of the study, the highest NI8 concentration had greater mortality (90.00 ± 3.18%) than the KUDSC strain (86.67 ± 3.60%) (Table 1). Fifth instar nymphs were highly susceptible to the KUDSC strain at all test concentrations. Significant differences in mortality were observed for 5th instars at 5 d (*F* = 33.13; df = 8, 809; *p* < 0.0001), day 10 (*F* = 24.01; df = 8, 809; *p* < 0.0001), and 20 d (*F* = 23.82; df = 8, 809; *p* < 0.0001) post-exposure (Table 1).

### 3.3. Lethal Concentration and Resistance Ratio of Life Stages of Kudzu Bug

The LC_50_ values (spores/mm^2^), as determined by probit analysis, indicated variability among life stages of kudzu bugs treated with both strains of *B. bassiana* (Table 2 and Table 3). For all life stages, the probit model produced a good fit of the data among the concentrations. Comparison of dose–mortality values using resistance ratios (RR_50_) was significantly different between life stages of kudzu bug for both strains of *B. bassiana*. The LC_50_ values showed that kudzu bug adults are more susceptible to both strains than any other life stage. The KUDSC strain was more pathogenic than NI8 at 10 d after exposure, but NI8 was significantly higher than KUDSC 20 d after exposure. Fifth instars were found to be significantly more tolerant to NI8 than all other life stages, with a RR_50_ 183-fold higher than adults, and 11-fold, 18-fold, and 6-fold less pathogenic when compared with 2nd, 3rd, and 4th instars, respectively (Table 3). No significant differences in pathogenicity were observed when nymphal stages were exposed to the KUDSC strain (Table 3).

## 4. Discussion

The data from these experiments suggest potential field usage of *B. bassiana* for control of kudzu bugs and their integration into pest management strategies, particularly in organic production systems where synthetic pesticides are not an option. This is unlikely to be successful in conventionally grown crops, where commercially available fungicides are known to suppress *B. bassiana* efficacy in the field [35] and have been reported to significantly decrease the number of sporulated kudzu bug cadavers observed in a Georgia soybean field [36]. Population dynamics of kudzu bugs in both kudzu and soybean are known to affect the levels of control achieved through application of insecticidal sprays [37], suggesting that similar follow-up studies should be done to investigate potential timing of fungal sprays both in crops and non-crop natural areas that act as population sources before kudzu bugs reach higher densities and become urban pests. The high susceptibility of all life stages of kudzu bugs observed in our study indicates that any stage can acquire lethal doses of conidia from direct spray, and a single spore/mm^2^ can kill adults regardless of the strain, which corroborated with data reported in another study [25] that evaluated mortality of young and old adults of kudzu bugs with two native and one commercial *B. bassiana* strains. Until today, no study has directly measured the pathogenicity of *B. bassiana* on nymphal stages of kudzu bugs; therefore, no comparable results were available for this part of the study.

While this study utilized only two strains of *B. bassiana*, recent work identified 42 isolates from soils in seven kudzu patches in North Carolina [38], suggesting that additional isolates could be considered for screening. Potential synergies exist with other biopesticides and biological control agents [39]. Studies indicate that host plant resistance also varies among legume species and soybean varieties [40], adding additional complexity to the system that could explain the frequent variation in fields.

While *B. bassiana* may not currently be a feasible component of large-scale commercial soybean production in the southeastern United States, biological control agents have additional roles alongside the assumed usages in organic crop production. Fungal biological control agents like *B. bassiana* could be proactively utilized when kudzu bugs and other insects form aggregations and become urban pests on structures to prevent human health hazards in the United States [19]. A similar strategy has been used in Argentina to control synthetic insecticide-resistant hemipteran pests that transmit human diseases [41]. Other potential uses may be to reduce populations of kudzu bugs in non-crop alternative hosts, similar to suggestions previously made for tarnished plant bugs (*Lygus* spp.) [36].

## Figures and Tables

**Figure 1 pathogens-11-01028-f001:**
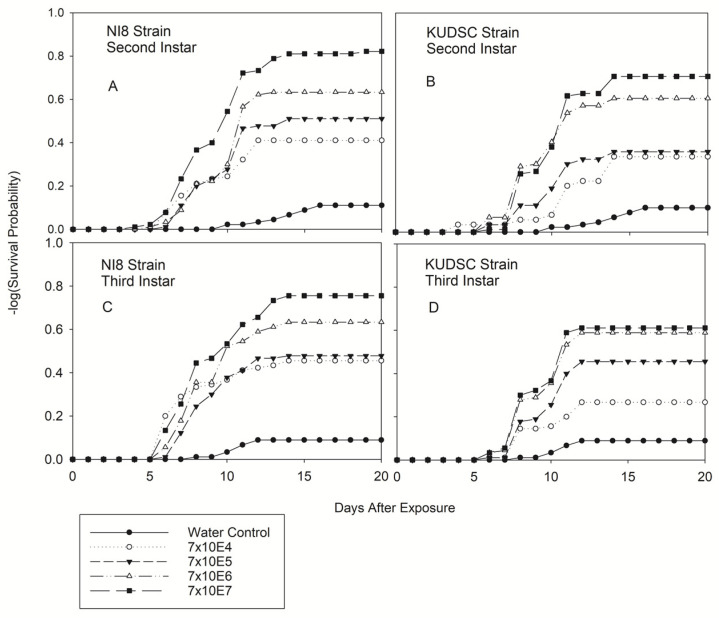
Survival estimates for second and third instar nymphs of kudzu bugs exposed to two native strains of *Beauveria bassiana* [NI8 (Mississippi Delta) (**A**,**C**) and KUDSC (South Carolina) (**B**,**D**)] at four concentrations (spores/mL). Survival probability at age x (*p* = 0.5, LIFETEST of Equality over Strata).

**Figure 2 pathogens-11-01028-f002:**
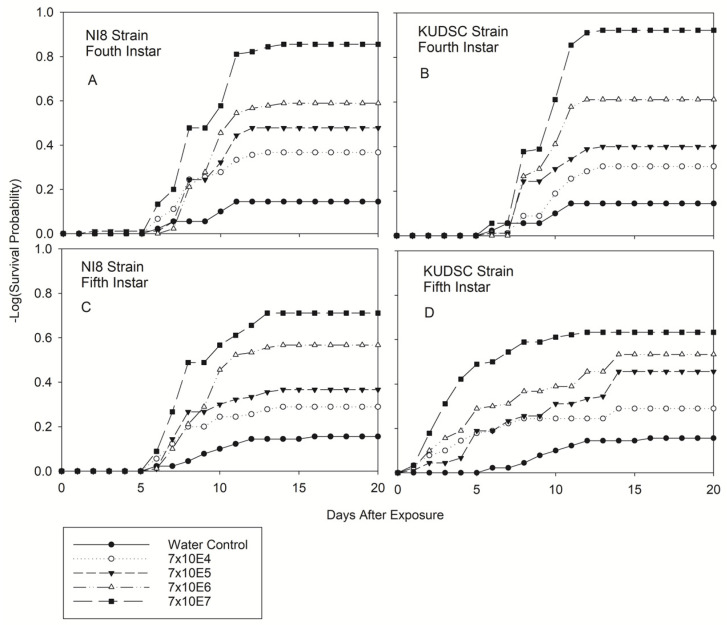
Product-limit survival estimates for fourth and fifth instar nymphs of kudzu bugs exposed to two native strains of *Beauveria bassiana* [NI8 (Mississippi Delta) and KUDSC (South Carolina)] at four concentrations (spores/mL) (**A**–**D**). Survival probability at age x (*p* = 0.5, LIFETEST of Equality over Strata).

**Figure 3 pathogens-11-01028-f003:**
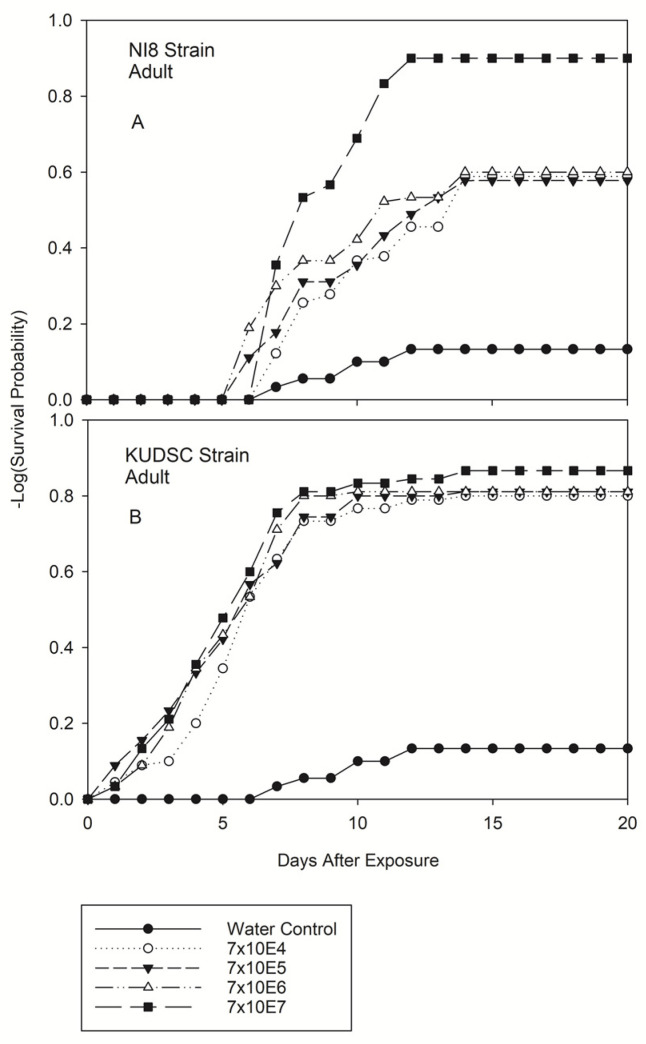
Product-limit survival estimates for adults of kudzu bugs exposed to two native strains of *Beauveria bassiana* [NI8 (Mississippi Delta) (**A**) and KUDSC (South Carolina) (**B**)] at four concentrations (spores/mL). Survival probability at age x (*p* = 0.5, LIFETEST of Equality over Strata).

**Table 1 pathogens-11-01028-t001:** Mortality of kudzu bug nymphs and adults following exposure to different concentrations of two *Beauveria bassiana* native strains from South Carolina (KUDSC) and Mississippi Delta (NI8).

Treatments ^1^ (Spores/mL)	Kudzu Bug Stages (% Mean)				
	2nd Instar	3rd Instar	4th Instar	5th Instar	Adult
**Day 10**					
Control	12.21 ± 3.54 de	10.0 ± 3.23 d	13.33 ± 3.6 c	13.33 ± 3.6 d	14.44 ± 3.72 d
7 × 10^4^-NI8	24.4 ± 4.63 bcde	36.7 ± 5.1 abc	26.66 ± 4.67 bc	24.44 ± 4.56 dc	38.89 ± 5.17 c
7 × 10^5^-NI8	27.82 ± 4.74 bcde	40.0 ± 5.21 ab	32.22 ± 4.95 bc	31.11 ± 4.91 dc	37.78 ± 5.14 c
7 × 10^6^-NI8	31.12 ± 4.93 cdb	53.3 ± 5.35 a	43.33 ± 5.25 ab	45.56 ± 5.28 abc	51.11 ± 5.30 bc
7 × 10^7^-NI8	56.70 ± 5.30 a	55.6 ± 5.2 a	56.67 ± 5.25 a	57.78 ± 5.24 a	68.89 ± 4.91 ab
7 × 10^4^-KUDSC	7.81 ± 2.80 e	15.6 ± 3.82 cd	22.22 ± 4.41 bc	31.11 ± 4.91 dc	76.67 ± 4.48 a
7 × 10^5^-KUDSC	20.00 ± 4.24 cde	27.8 ± 4.76 bcd	27.78 ± 4.75 bc	26.67 ± 4.69 dc	80.00 ± 4.24 a
7 × 10^6^-KUDSC	42.23 ± 5.21 ab	35.6 ± 5.14 abc	42.22 ± 5.24 ab	38.89 ± 5.17 bc	81.11 ± 4.15 a
7 × 10^7^-KUDSC	40.05 ± 5.28 abc	53.3 ± 5.3 a	63.33 ± 5.11 a	64.44 ± 4.5 a	86.67 ± 3.6 a
**Day 20**					
Control	21.12 ± 4.39 f	17.80 ± 4.12 d	21.11 ± 4.33 d	16.67 ± 2.53 e	15.56 ± 3.84 e
7 × 10^4^-NI8	41.16 ± 5.25 edf	46.75 ± 5.3 bc	35.56 ± 5.07 cd	28.89 ± 4.8 ed	62.22 ± 5.14 dc
7 × 10^5^-NI8	51.13 ± 5.33 edc	50.03 ± 5.3 b	48.89 ± 5.3 bc	37.78 ± 5.14 edc	56.67 ± 5.25 d
7 × 10^6^-NI8	64.40 ± 5.10 abc	64.41 ± 4.51 ab	58.89 ± 5.22 b	56.67 ± 5.25 abc	70.00 ± 4.85 bcd
7 × 10^7^-NI8	84.41 ± 3.83 a	76.72 ± 4.8 a	85.56 ± 3.73 a	72.22 ± 5.28 a	90.00 ± 3.18 a
7 × 10^4^-KUDSC	37.78 ± 5.17 ef	25.63 ± 4.68 cd	33.33 ± 4.5 cd	45.56 ± 5.28 bdc	80.00 ± 4.23 abc
7 × 10^5^-KUDSC	37.03 ± 5.12 ef	47.83 ± 5.33 b	37.78 ± 5.14 cd	31.11 ± 4.9 ed	81.11 ± 4.15 abc
7 × 10^6^-KUDSC	62.20 ± 5.15 edc	64.40 ± 5.1 ab	63.33 ± 5.11 b	53.33 ± 5.29 abc	81.11 ± 4.45 abc
7 × 10^7^-KUDSC	73.33 ± 4.70 ab	85.65 ± 3.71 a	95.56 ± 2.18 a	66.67 ± 4.5 ab	86.67 ± 3.6 ab

Means ± SD. Followed by the same letter in each column are not significantly different (*p* < 0.05 Tukey test).

**Table 2 pathogens-11-01028-t002:** Lethal mortality-response (LC_50_) of different stages of kudzu bugs treated with *Beauveria bassiana*, Mississippi Delta native NI8 strains.

Stage	Concentration Response (Spores/mm^2^)							
	*n*	Slope ± SE	LC_50_ (95% CI)	Probit Trend				RR_50_ (95% CI) *
				Test for Slope		Test for GoF		
				X^2^	*p* > X^2^	X^2^	*p* > X^2^	
**10 Days**								
2nd Instar	450	0.18 ± 0.07	789 (77–3.1 × 10^26^)	5.3	0.0214	1.89	0.0415	17 (1–309)
3rd Instar	450	0.09 ± 0.03	199 (24–7.3 × 10^4^)	8.51	0.0035	0.68	0.7436	3 (0.13–50)
4th Instar	450	0.15 ± 0.04	480 (101–1.0 × 10^4^)	14.18	0.0002	1.1	0.3551	8 (0.61–110)
5th Instar	450	0.18 ± 0.04	307 (85–2.4 ×10^3^)	17.29	0.0001	1.44	0.9265	6 (0.57–73)
Adult	450	0.15 ± 0.04	73 (5–4.5 × 10^3^)	9.3	0.0023	1.75	0.0568	1
**20 Days**								
2nd Instar	450	0.22 ± 0.04	13 (3–36)	28.65	0.0001	1.22	0.2680	16 (0.46–588)
3rd Instar	450	0.15 ± 0.03	11 (2–45)	18.46	0.0001	1.48	0.2265	10 (0.22–429)
4th Instar	450	0.25 ± 0.05	20 (6–51)	28.03	0.0001	1.23	0.2624	32 (0.94–1096)
5th Instar	450	0.17 ± 0.03	133 (33–879)	18.97	0.0001	1.24	0.2532	183 (5–6494)
Adult	450	0.54 ± 0.05	1 (0.0005–6)	10.09	0.0015	1.93	0.0363	1

* Differences among RR_50_ values are significant if 95% CI does not include 1.0. RR_50_ compared to LC_50_s among the lowest LC_50_ as a control.

**Table 3 pathogens-11-01028-t003:** Lethal mortality–response (LC_50_) of *Kudzu bugs* treated with *Beauveria bassiana*, South Carolina native KUDSC strain.

Stage	Concentration Response (Spores/mm^2^)							
	*n*	Slope ± SE	LC_50_ (95% CI)	Probit Trend				RR_50_ (95% CI) *
				Test for Slope		Test for GoF		
				X^2^	*p* > X^2^	X^2^	*p* > X^2^	
**10 Days**								
2nd Instar	450	0.21 ± 0.04	1051 (286–12,053)	22.54	0.0001	1.52	0.1249	438 (71–2694)
3rd Instar	450	0.19 ± 0.04	464 (140–3705)	19.93	0.0001	1.04	0.4043	184 (35–981)
4th Instar	450	0.23 ± 0.05	205 (70–864)	17.42	0.0001	1.11	0.3462	67 (15–360)
5th Instar	450	0.22 ± 0.05	182 (61–748)	19.56	0.0001	0.83	0.5931	58 (13–271)
Adult	450	0.23 ± 0.03	3 (1–7)	47.51	0.0001	1.31	0.2159	1
**20 Days**								
2nd Instar	450	0.21 ± 0.04	49 (12–164)	19.66	0.0001	0.67	0.7573	25 (1.32–482)
3rd Instar	450	0.30 ± 0.04	17 (7–36)	44.11	0.0001	0.84	0.5889	19 (1.26–284)
4th Instar	450	0.52 ± 0.15	28 (2–65)	11.28	0.0008	1.84	0.0479	32 (1.74–580)
5th Instar	450	0.21 ± 0.04	82 (27–297)	25.38	0.0001	1.51	0.1267	68 (4–1117)
Adult	450	0.21 ± 0.06	1.14 (0.006–7)	11.62	0.0007	3.22	0.0004	1

* Differences among RR_50_ values are significant if 95% CI do not include 1.0. RR_50_ compare the LC_50_s among the lowest LC_50_ as a control.

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
