# Peer review of "Susceptibility of Different Life Stages of Kudzu Bug Megacopta cribraria (F.) (Hemiptera: Plataspidae) to Two Different Native Strains of Beauveria bassiana"

_pathogens, 2022, doi:10.3390/pathogens11091028_

Round 1
Reviewer 1 Report
A nice manuscript. Edits provided on the marked and scanned copy.

Reviewer 2 Report
The set goals on Megacopta cribraria have been achieved. The exposition is accurate and comprehensive. Further investigation of the effects of Beauveria on antagonistic insects is suggested before moving to direct action in crops.
Author Response
No comments or revisions from reviewer # 2 were attached.
Reviewer 3 Report
This manuscript presents the time-concentration-mortality effects of two native B. bassiana isolates on nymphs and adults of Kudzu bug, a soybean pest introduced to North America. All experiments were well done. All bioassay data were well analyzed, presented and interpreted. I recommend the manuscript to be accepted by the editor after a minor revision.
I suggest the authors to consider time-dose-mortality modeling analysis (see J. Invertebr. Pathol., 1998, 72: 246–251) for better differentiation of time and concentration effects and possible interaction if possible.
Author Response
Dear Reviewer
A follow up paper with time-dose-mortality will be coming soon, testing the native strain KUDSC and two commercial strain GHA (B. bassiana) and NoVil (M. robertsii) under lab and greenhouse conditions before moving to field trials on soybean crops. So, we prefer to keep the analysis as it is.